# *‘Changing the Focus’*: Co-Design of a Novel Approach for Engaging People with Dementia in Physical Activity

**DOI:** 10.3390/nursrep15010002

**Published:** 2024-12-24

**Authors:** Claudia Meyer, Den-Ching A. Lee, Michele Callisaya, Morag E. Taylor, Katherine Lawler, Pazit Levinger, Susan Hunter, Dawn C. Mackey, Elissa Burton, Natasha Brusco, Terry Haines, Christina L. Ekegren, Amelia Crabtree, Keith D. Hill

**Affiliations:** 1Bolton Clarke Research Institute, Forest Hill, VIC 3131, Australia; 2Rehabilitation, Ageing and Independent Living (RAIL) Research Centre, Monash University, Frankston, VIC 3199, Australia; angel.lee@monash.edu (D.-C.A.L.); natasha.brusco@monash.edu (N.B.); christina.ekegren@monash.edu (C.L.E.); keith.hill@monash.edu (K.D.H.); 3College of Nursing and Health Sciences, Flinders University, Adelaide, SA 5000, Australia; 4Centre for Health Communication and Participation, La Trobe University, Bundoora, VIC 3083, Australia; 5National Centre for Healthy Ageing (NCHA), Monash University (Peninsula Campus) and Peninsula Health, Frankston, VIC 3199, Australia; michele.callisaya@monash.edu (M.C.); terry.haines@monash.edu (T.H.); 6Peninsula Clinical School, Central Clinical School, Monash University, Frankston, VIC 3199, Australia; 7Menzies Institute for Medical Research, University of Tasmania, Hobart, TAS 7005, Australia; 8Falls, Balance and Injury Research Centre, Neuroscience Research Australia, Randwick, NSW 2031, Australia; morag.taylor@unsw.edu.au; 9Physiotherapy, School of Health Sciences, Faculty of Medicine and Health, UNSW Sydney, Sydney, NSW 2033, Australia; 10School of Allied Health, Human Services and Sport, La Trobe University, Bundoora, VIC 3083, Australia; kate.lawler@latrobe.edu.au; 11Wicking Dementia Research and Education Centre, College of Health and Medicine, University of Tasmania, Hobart, TAS 7000, Australia; 12National Ageing Research Institute, Parkville, VIC 3052, Australia; p.levinger@nari.edu.au; 13School of Physical Therapy, University of Western Ontario, London, ON N6A 0E5, Canada; susan.hunter@uwo.ca; 14Department of Biomedical Physiology and Kinesiology, Simon Fraser University, Burnaby, BC V5A 1S6, Canada; dmackey@sfu.ca; 15Curtin School of Allied Health, Faculty of Health Sciences, Curtin University, Perth, WA 6000, Australia; e.burton@curtin.edu.au; 16enAble Institute, Faculty of Health Sciences, Curtin University, Perth, WA 6000, Australia; 17School of Primary and Allied Health Care, Monash University (Peninsula Campus), Frankston, VIC 3199, Australia; 18Aged & Rehabilitation Division, Monash Health, Clayton, VIC 3168, Australia; amelia.crabtree@monashhealth.org; 19School of Clinical Sciences at Monash Health, Faculty of Medicine, Nursing and Health Sciences, Monash University, Clayton, VIC 3168, Australia

**Keywords:** dementia, physical activity, community, co-design, nursing

## Abstract

Background: Promoting physical activity among people living with dementia is critical to maximise physical, cognitive and social benefits; yet the lack of knowledge, skills and confidence among health professionals, informal care partners and people with dementia deters participation. As the initial phase of a larger feasibility study, co-design was employed to develop a new model of community care, *‘Changing the Focus’,* to facilitate the physical activity participation of older people living with mild dementia. Methods: Co-design methodology was utilised with nine stakeholders (with experience in referring to or providing physical activity programs and/or contributing to policy and program planning) over three workshops plus individual interviews with four care partners of people with dementia. Insights were gathered on the physical activity for people with mild dementia, referral pathways were explored and ‘personas’ were developed and refined. Materials and resources to support exercise providers and referrers to work effectively with people with mild dementia were finalised. Results: Three ‘personas’ emerged from the co-design sessions, aligned with stages of behaviour change: (1) hesitant to engage; (2) preparing to engage; and (3) actively engaged. Referral pathway discussions identified challenges related to limited resources, limited knowledge, access constraints and individual factors. Opportunities were classified as using champions, streamlining processes, recognising triggers for disengagement, influencing beliefs and attitudes, and means of communication. Conclusion: This study captured the views of physical activity referrers and providers and informal care partners in an inclusive and iterative manner. The use of co-design ensured a robust approach to facilitating participation in formal and informal physical activity options for people living with mild dementia. This study has provided the necessary framework from which to develop and test training and resources for the next stage of intervention (a feasibility trial) to improve physical activity participation for people with dementia.

## 1. Introduction

Globally, approximately 50 million people live with dementia and are directly impacted by the cognitive and behavioural changes attributed to disease processes [1]. Concomitantly, physical changes result in reduced independence, physical function (including balance and mobility), and community participation, and increased falls [2,3]. These changes, coupled with low levels of physical activity among older people in general [4] and worsened through dementia disease progression [5], can impact wellbeing and quality of life, including the worsening of cognitive and physical changes. However, these losses can be tempered as evidence exists for physical activity programs that can be safely implemented with people with dementia, with similar outcomes for the general older population, such as improving function and physical fitness [6,7] and slowing cognitive decline [8]. Numerous barriers to physical activity for people with dementia exist, such as difficulties with motivation and apathy, physical health factors, and feelings of not being understood or being forced to exercise [9]. Further, costs and strict schedules and a lack of accommodation to preferences have been found to deter people living with dementia engaging in physical activity programs [10]. There appears to be no current systematic approach, using shared decision making, for identifying needs and preferences, assessing and promoting physical activity, for people diagnosed with dementia.

Health and physical activity professionals, including doctors, community-based nurses, and allied health professionals, who work with people living with dementia are well placed to either refer or accept referrals for physical activity programs. Nonetheless, a noticeable gap exists in understanding dementia, recognising the advantages and health benefits of physical activity [11], and having the confidence to effectively work with this clinical group [12]. For people with dementia, active involvement in physical activities can be significantly improved if they perceive physical activity as meaningful and recognise the tangible benefits. This can be fostered through collaborative care partnerships that allow for changes in the person’s cognition over time [13].

There exists a pressing need to support the involvement of people living with dementia in physical activity programs. However, the complexity of health and aged care systems, including community services, makes it challenging to establish a robust approach to promoting participation in physical activity. Leisure spaces, such as community centres and fitness areas, are well placed to be safe and accessible spaces, providing meaningful activity for people with dementia [14] but potentially are not yet adequately equipped to do so. The application of co-design methodology is useful to engage those who use or are part of these systems, ensuring program relevance and pragmatism [15]. Co-design methodology involves meaningful end-user engagement to better understand needs and preferences [16], with both benefits and limitations of including people with dementia in this process [17].

The aims of this study were to identify stakeholders’ perspectives on factors that would influence participation in a novel model of community care, *‘Changing the Focus’*, and to develop recommended strategies to be incorporated into a planned feasibility trial. This innovative approach aimed to facilitate uptake and sustained physical activity participation specifically for older people living with mild dementia.

## 2. Methods

### 2.1. Study Design

This study, using co-design methodology through working together collaboratively to solve problems, was the first component of a larger feasibility study [18], conducted in the Frankston/Mornington Peninsula region of Victoria, Australia.

The co-design process was structured on the Double Diamond design framework [19] and work by Boyd et al. [20], adapted by the Bolton Clarke Research Institute [21], providing a robust foundation for collaborative work involved in this study (Figure 1).

Phase 1 of this study (‘discover’) aimed to gather information and insights from two groups of stakeholders via online workshops and individual interviews, and Phase 2 (‘define’ and ‘develop’) aimed to synthesise Phase 1 findings to turn ideas into action via persona development.

The final component of ‘deliver’ will be achieved via a feasibility trial [18], which will be informed by this co-design work.

### 2.2. Methodology

#### 2.2.1. Phase 1 (Discover): Gathering Information and Insights 

Information and insights were planned to be sought from two separate stakeholder groups: people planning, promoting, referring to, or delivering physical activity programs for people living with dementia (Group 1); and people living with mild dementia and/or their care partners (Group 2).

##### Participant Recruitment and Consent Process

Group 1: Individuals and representatives from local organisations and the local government, specifically those involved in providing or delivering physical activity, making referrals, and/or contributing to policy and program planning, were invited to participate. Inclusion criteria were working in the Frankston and Mornington Peninsula region and holding a role that supports, advises, influences or in any other way supports older people, including those living with mild dementia, in engaging in physical activity.

Invitations were extended to potential participants via email, consisting of an explanatory statement, a consent form, and a study flyer, sent by the research team to the local government (Frankston City Council, Mornington Peninsula Shire), primary care services, community aged care service providers, gymnasiums, physiotherapy/exercise physiology private practice, peak bodies (e.g., Dementia Australia) or researchers’ networks in the local area. All interested individuals were asked to complete a written consent form.

Group 2: Older people (≥60 years) living with mild dementia (Mini Mental State Examination (MMSE) 18–23 inclusive if no medical diagnosis of dementia) and/or their unpaid informal care partners were invited to participate. To be included, people with dementia needed to be able to provide consent for themselves (evaluated using an Evaluation of Capacity to Consent [22], further details below), and care partners needed to have current or previous caregiving experience of people with any stage of dementia. People were invited to participate via promotional material through local councils, Umbrella Dementia Cafés (informal peer support groups designed for families and people living with dementia to improve social health and wellbeing), and researcher and community aged care provider networks.

Potential participants were provided with an Explanatory Statement, giving them time to read and ask questions as needed, and to sign the consent form. This included care partners and people living with dementia interested in participating. For any person living with mild dementia interested in participating, ability to provide consent for themselves was ascertained by providing them with the Explanatory Statement and undertaking an Evaluation of Capacity to Consent. The researchers had knowledge and experience in the use of the checklist and interpretation of the results. The checklist includes asking the person with dementia to clarify, in their own words, what involvement will entail, as well as the risks of involvement (rather than only being asked to respond yes or no to the questions).

##### Data Collection and Analysis

Group 1: Three 2 h online workshops were conducted between May and July 2023, facilitated by CM with experience in co-design methodology and supported by team researchers KH and DL. Participants were asked to commit to all three workshops as able. Notes were generated from each of the sessions, aided by audio-recordings via the Zoom platform. The project team revised notes between sessions, developing draft, and then revised personas.

The objectives of the first session were to (1) explore perceived likes and dislikes related to physical activity for older people generally and people with dementia specifically; (2) explore perceived reasons for people with dementia starting, stopping and/or continuing with physical activity; and (3) identify physical activity options in the Frankston/Mornington Peninsula region and whether these options may be appropriate, acceptable and feasible for people with dementia.

The objectives of the second session were to (1) explore referral pathways for physical activity participation for people living with mild dementia; (2) understand opportunities and barriers, including areas of challenge for referral and acceptance of referral; and (3) identify key characteristics and skills of physical activity providers that support engagement.

The objectives of the third session were to explore each of the three identified personas for the material and resource requirements to support physical activity providers and referrers in encouraging the participation of people with mild dementia in physical activity. In addition, locally relevant referral pathways were confirmed and insights for training resources for the intervention were gathered.

Group 2: To accommodate diverse communication needs and preferences, participants were offered individual face-to-face or online interviews over two sessions as an alternative to joining a group workshop. Interviews were audio-recorded, with transcripts generated via Microsoft Teams software (version 24295.605.3225.8804) for online interviews and manually checked for accuracy. Interview questions included the following:Why do people with mild dementia participate in physical activity? What do they like about physical activity? What don’t they like about physical activity? Why?For the person you are caring for, what type of physical activity has been tried? How did they get started? What has enabled them to continue with it? Why did they stop?Please share your insights on whether these community-based physical activity options would be feasible and acceptable (for example, general group classes, water exercise, individually tailored rehabilitation programs, gentle exercise classes, leisure options, dancing, strengthening, walking, seniors’ groups, community houses, community gardens).

Interview transcripts were reviewed, with key categories identified and triangulated with Group 1 data for persona development (see below for more detail). Personas were sent to the interviewees for verbal or written feedback. A follow-up phone call was arranged to discuss feedback if the participants desired.

#### 2.2.2. Phase 2 (Define and Develop): Synthesising Findings and Turning Ideas into Action

Phase 2 incorporated persona development, a well-established co-design technique utilised in software design, proven to be invaluable in comprehending user needs and predicting how ‘personas’ will interact with a system or program [23]. Personas are fictional characters, which are created based upon research to represent the different user types that might use the service, product, site, or brand in a similar way. Creating personas helps designers to understand users’ needs, experiences, behaviours, and goals. The qualitative data from Phase 1 workshops and interviews guided persona development, through conceptualising relevant categories, prioritising key information and validating the personas by double-checking with participants, a similar process to content analysis [23]. In the context of physical activity participation for people with dementia, persona development becomes helpful as it gives form to the diverse stakeholder perspectives inherent in the co-design process.

Preliminary personas were developed from the first Group 1 workshop and the Group 2 interviews (Phase 1). Personas were refined after the second Group 1 workshop and Group 2 feedback, with final personas endorsed during the third Group 1 workshop. The personas were subsequently embedded within the decision support tool designed for the 12-month intervention, assisting the study therapists to engage appropriate strategies for motivation in physical activity [18].

## 3. Results

Twelve people participated in the co-design workshops, representing universities/research institutions (n = 3, unit director, senior research fellow, and research fellow), community health centres (n = 2, centre managers), local council (n = 2, Positive Ageing teams), leisure centres (n = 1, exercise physiologist), community care providers (n = 1, physiotherapist), private practice (n = 1, physiotherapist), health services (n = 1, geriatrician), and Dementia Australia (n = 1, social support officer).

Four informal care partners of people with dementia were interviewed via Microsoft Teams or face to face in their homes at a mutually convenient time. Care partners were of Anglo-Saxon descent and of mid socio-economic status. The recruitment of people with dementia was through local community organisations, aged care providers, and avenues known to the research team; yet, despite these attempts, no participants with dementia were recruited.

### 3.1. Phase 1 (Discover): Gathering Information and Insights

#### 3.1.1. Insights from Physical Activity Referrers and/or Providers (Group 1)

***Session no. 1*** identified factors relating to physical activity preferences and dislikes, reasons for starting, continuing, or ceasing physical activity, and the feasibility and acceptability of physical activity options for people with dementia from the perspective of workshop participants. The findings have been summarised and grouped according to the need for physical activity delivery via the following: (1) an individual setting; (2) a group setting that suits people who are less confident and social; and (3) a group setting for people who are more confident and social. These findings informed the first shaping of personas (see Table 1). In addition, there were several points raised from the perspective of workshop participants that were relevant across all groupings:Physical activity needs a purpose relevant to the individual and needs to be fun;A welcoming and inclusive environment is preferred;The initial experience must be positive, with care taken to minimise pain and fatigue;One must aim for mastery, relating goals to independence, quality of life and doing what they love;It is necessary to understand variable funding options to assist with costs;There is an option to have a ‘try before buying’.

During this session, participants were presented with a directory of over 200 physical activity, social and community groups available in Frankston and the Mornington Peninsula region [24]. This free resource was deemed suitable by participants for identifying local options, with each option requiring exploration for individual appropriateness and acceptability. In addition, skills, knowledge and personal attributes of health professionals and physical activity leaders were noted for follow-up in session no. 2.

Session no. 2 explored existing and potential referral pathways for physical activity, including barriers and challenges, and identified key characteristics and skills required to support participation in physical activity by people with mild dementia.

A summary of referral pathways to physical activity options included the following:Private practice (e.g., general practitioners/nurse practitioners, physiotherapists);Hospital-based (e.g., medical specialists, inpatient clinicians, outpatient clinicians, including cognitive assessment clinics);Community services (e.g., council services, recreation centres, service directory);Self-referral (e.g., self, family members/care partners, friends/peers).

Challenges were related to (1) the limited resources of health and care professionals, (2) the limited knowledge of health and care professionals, people living with dementia and their care partners, (3) access issues and (4) person-related factors. Opportunities included (1) physical activity champions, (2) streamlining processes, (3) the recognition of triggers for reducing capability and motivation (i.e., noting functional decline that may trigger a reduction in physical activity levels), (4) beliefs and attitudes of health and care professionals, people living with mild dementia and their care partners, and (5) communication in the broader public domain. Further detail is outlined in Table 2.

#### 3.1.2. Insights from Care Partners of People Living with Dementia

Care partners shared their perspectives on engaging people with dementia in physical activity. The first category to emerge was ‘a hesitance to engage in physical activity’. For some people with dementia, there was a denial of any cognitive or physical change whether that be due to the dementia process or ageing generally. Numerous reasons were given for why they did not wish to engage with physical activity such as ‘*too tired’* or *‘can’t be bothered’.* Care partners mentioned that the person with dementia may be aware of physical and/or cognitive deterioration, which caused embarrassment and a loss of confidence, with the potential of exposing their level of frailty and possibly a loss of control when participating in physical activities. Further, hesitance to engage was related to prohibitive physical (e.g., transport) and financial (e.g., fee-for-service costs) accessibility.

The second category to emerge was ‘finding the motivation to participate’. Motivation was enhanced by linking physical activity with a favourite activity, or previous work/home role, of the person with dementia as an incentive. Motivation was also found when the person with dementia was able to teach others—to share their physical activity skills with peers or to assist with upskilling physical activity leaders. Being offered reassurance by the physical activity leader, with a focus on physical and cognitive abilities, and ensuring physical activity starts at the level of ‘just right’ was critical; but it was noted that clubs and groups can be highly variable regarding inclusion and willingness to adjust to the needs and preferences of the person with dementia.

The third, and final, category to emerge was ‘maintaining consistency in physical activity participation’. Having a care partner/friend to provide physical and emotional support was beneficial but must be acknowledged as often time-consuming and challenging for them, while, alternatively, having paid care partners who are familiar to, and well liked by, the person with dementia can be a source of support (i.e., during respite care periods). A physical activity routine, with a regular day/time, can be beneficial to ensure continuity and ongoing momentum, while some ‘bending of the rules’ for a favourite sport or hobby as physical and/or cognitive deterioration occurred was appreciated (e.g., easing the rigidity of golf rules to encourage participation).

### 3.2. Phase 2 (Define and Develop): Synthesising Findings and Turning Ideas into Action

#### 3.2.1. Persona Refinement

Preliminary personas were refined based on findings from session no. 1 and 2 with stakeholders and insights from care partners as articulated above. Feedback was gathered in session 3 with Group 1 stakeholders and written/verbal feedback from care partners on the revised personas. Only small adjustments were needed such as rather than personas being based on individual or group-based preferences for physical activity (considered too restrictive), they were modified to be based around preparedness for involvement in physical activity by the person living with mild dementia. Partly, this was due to people living with dementia likely needing differing advice and support based on their readiness to change regardless of their preference for the physical activity mode. Initially, ‘reluctance to exercise’ was used, with this being viewed as a negative term and hence changed to ‘hesitance to exercise’. The three personas were endorsed as outlined below (with supporting quotes), noting that individuals can move between them:Hesitant to engage—This persona represented reduced motivation to engage in physical activity. Some reasons for avoiding physical activity related to denial and embarrassment, while others related to a lack of interest (apathy) or access to appropriate activities. This persona may frustrate informal care partners and may benefit from linking physical activity indirectly to other activities or with peers (e.g., walking to a coffee shop).“*he completely denied that he had any memory cognitive issues and thought he was quite capable of making his own decisions about what he wanted to do…and if he didn’t want to get up and go out somewhere, then why should he?*” (care partner).Preparing to engage—Finding motivation is key for this persona, linking to previous positive experiences. Assisting with logistical requirements is critical given that cognitive impairment may make initiating and planning physical activity difficult. A recognition of reducing confidence and/or changing physical or mental health needs throughout disease progression are important for physical activity providers for people linked to this persona.“*I’m sure that the carer is key, but you’ve also got whoever is the person conducting the activity, they certainly need to know the person and be able to gauge the exercise activity to their level of acceptance and capacity*” (care partner).Actively engaged—linked with the active and maintenance stages of behaviour change. This persona may represent people with positive health-seeking behaviours and having a sense of fun and purpose in physical activities that they currently undertake. However, it can also include people who are doing a small amount of physical activity but not meeting recommended levels. Critical to this persona is the ability of health professionals to respond promptly to changes in ability or motivation, through program adaptation or change in activity, to prevent a loss of momentum.“*it’s difficult when hospital discharge is all about just getting people back home…it’s a subtle skill that’s needed to elicit usual function that is beyond what is the bare minimum to safely get someone back home. It should be gold standard to get people back to where they were a year ago, not just two weeks ago*” (stakeholder).

#### 3.2.2. Professional Training and Resources

Discussions about referral pathways highlighted the need for change in knowledge, skills, and attitudes for: (1) people referring to physical activities; (2) people accepting referrals and delivering physical activities; and (3) people living with mild dementia and their care partners who would be participating in physical activities.

Informed by the co-design work, a self-paced, 3 h e-training program was developed as part of the planned ‘*Changing the Focus’* feasibility trial [18]. This resource is freely available (through the project team) for referrers (those involved in the diagnosis and/or care of people living with mild dementia) and physical activity program providers looking to enhance their understanding of strategies for supporting people with dementia in physical activity programs. The training program (see Table 3 for details) includes the following topics:Welcome and an overview of the *‘Changing the Focus’* program;Module 1: An introduction to dementia;Module 2: Changes in physical function and benefits of physical activity for people with dementia;Module 3: Engaging people with dementia in exercise programs and/or physical activities, including communication strategies;Module 4: Environmental design factors for dementia-friendly exercise facilities or environments;Module 5: Referral, funding options and resources for people with dementia

In addition, drawing on co-design findings, a decision-making support tool was developed (refer to protocol paper [18]). The support tool incorporates a framework for a discussion of the following points informing the choice/s of physical activity to be trialled to increase the physical activity participation of people living with mild dementia:Current physical activity (including a measure of minutes per week), and levels of enjoyment/commitment;Physical performance measures of balance, strength and fitness (identifying domains outside of age-related norms that might be considered to be a target for the physical activity program);Cognitive status and its potential impact for safe participation in some forms of physical activity;Concurrent health problems that may impact ability or choice of activities;Personal preferences;Transport and funding options/restrictions;Opportunity and motivating factors for physical activity;Physical activity options to explore in the local area;Goals for the program, and target duration of physical activity toward the World Health Organisation guideline of 150 min/week of moderate to vigorous activity [25].

## 4. Discussion

This study aimed to identify stakeholders’ perspectives on factors that would facilitate the uptake, and influence the participation, of sustained physical activity among older people living with mild dementia. This study has shown that physical activity options should be both flexible and pragmatic to meet the needs and preferences of people living with dementia and their care partners. Three personas—hesitant to engage, preparing to engage and actively engaged—were developed to guide a shared a decision-making approach for use within the forthcoming feasibility trial, supported by the training and resources [18].

This approach was critical to enhancing opportunity and motivation for engaging people with dementia in existing physical activity programs, rather than developing new programs, and the results are being incorporated into the approach being used by therapists in our feasibility trial and will inform some future training and resources. Evidence exists that physical activity can improve physical, cognitive and mental health outcomes for people living with dementia [6,7] particularly when introduced early in the disease progression. However, translating evidence into practice remains challenging and can take years to be fully integrated into policy and practice [26]. To bridge the gap between research and practice, it is crucial to engage key stakeholders and create pathways for collaboration. The methodology used in this study facilitated the engagement of health professionals and care partners of people living with dementia, providing a safe space for diverse views to be heard and discussed and valuing individual and collective perspectives [27]. Co-design methodology can have its challenges with potential power imbalance, information overload and difficulty interjecting and being heard [28]. The workshop with representatives from the local council, leisure centres, community health centres, community care providers, private practice, health services and Dementia Australia in one group, and care partners in another group (using individual interviews), ensured that information was collected concurrently but avoided some of these challenges.

The World Health Organisation calls on all people across the life course to undertake regular physical activity and to undertake 150 min/week of moderate to vigorous activity [29]. Within Australia, only 25% of older people meet physical activity guidelines [30], with people living with dementia even less active [5]. The three identified personas—hesitant to engage, preparing to engage, and actively engaged –provide a useful framework for guiding behavioural and motivational strategies to engage people living with mild dementia in physical activity, given that they align with the stages of behaviour change (pre-contemplation, contemplation, preparation, action and maintenance) as outlined by Prochaska et al. [31]. It is important to recognise that individuals can move between these personas and may exhibit characteristics from one or more at any given time. Maintaining a flexible approach and regular monitoring would ensure that interventions are tailored to meet the evolving needs and preferences of everyone, fostering more meaningful and sustained participation in physical activity programs.

The first persona of ‘hesitant to engage’ links with the precontemplation stage of behaviour change, where people are not intending to make any changes in behaviour soon. There is a focus on consciousness-raising for this persona, gentle and persistent support from trusted sources, and gathering information that may be of use to shift behaviour [31]. Dementia-friendly communities attempt to reduce stigma related to a loss of community connection and are a potential avenue for empowering and including people living with dementia in preferred community-based activities [32].

The second persona of ‘preparing to engage’ links with both the contemplation and preparatory stages of behaviour change, whereby there is an awareness of the need to change and to formulate an action plan. In this persona, people may already be participating in some, but limited, physical activities, which may provide a useful basis from which to build greater physical activity engagement. This persona may benefit from physical activity options undertaken with their care partner and/or intergenerational options via a buddy approach to support their changing needs [33,34]. Health and care interventions can often be action-oriented, but cognisance is needed to ensure people are encouraged gently in this phase [35].

The third persona of ‘actively engaged’ links with the action and maintenance stages of behaviour change. In this stage, modifications are made to sustain physical activity, with a particular emphasis on monitoring factors such as declining cognitive or physical performance, or periods of ill health, to minimise the risk of stopping or reducing physical activity participation gains. Reminders and routines may be of benefit here to maintain engagement. Providing choice in this stage can increase motivation [31], being mindful to use a shared decision-making approach [36].

Health professionals and physical activity providers along the referral pathway, including both those who refer to services and those who receive the referral, are critical to the successful uptake of physical activity among people living with dementia. It is necessary to build relationships between health professionals, physical activity providers, care partners and people living with dementia, ensuring a person-centred approach to understanding values and preferences [37]. Participants in this study highlighted challenges such as limited knowledge and resources of health professionals and access issues facing people with dementia. Dementia-friendly initiatives acknowledge the need for strong interdisciplinary collaboration and creating positive conceptualisations of dementia rather than reinforcing negative stereotypes [38], while dementia-enabling environments facilitate the creation of supportive environments, providing practical information for optimal care environments [39]. Breaking down dementia-related stigma through education and training is seen as critical to change attitudes and encourage help-seeking behaviour [40]. These solutions are applicable for health professionals at either end of the physical activity referral pathway.

Care partners in this study highlighted that the relationship between the care partner and the person with dementia, as it relates to physical activity, is complex. Increasing physical activity levels in the person with dementia can have a positive effect on care partners, whereby their physical and emotional support can encourage participation, yet for some care partners, this additional requirement to support physical activity participation for the person they provide care for may also increase their workload [41]. It has been noted in the literature that it is more than just the increased time and effort required for the physical activity but also the type and style of the activity, for which care partners would like their views heard [42]. Social connectedness was also identified as a motivator to participation in physical activity, which is supported by the literature [43]. In testing our approach, it is likely that shared decision making will be beneficial, whereby needs and preferences can be established and choices discussed between health and care professionals, care partners and the person living with dementia [36]. Involvement in decision making is critical to giving expression to a person’s unique identity, but, due to the differing nature of dementia presentation, it is also relational and changes over time [44]. Ongoing adjustments to physical activity choices, with the support of health and care professionals and care partners, is required as physical and cognitive ability, needs and preferences, and social situations change, with potential movement between personas. This will be explored in the forthcoming feasibility study to ultimately determine the program effectiveness and cost-effectiveness of the ‘*Changing the Focus’* approach.

The co-design methodology employed for this study allowed the voices of multiple stakeholders including health professionals, service managers, physical activity providers, local council representatives, and care partners of people with dementia to be heard. However, not all potential referrers were represented (such as community-based nurses), and a limitation of the study was the inability to recruit people living with dementia in the required timeframe (people with dementia will be interviewed as part of the feasibility study). As identified by Lord et al. [28], it is possible that memory problems reduce people’s confidence to participate in research, particularly when substantial information will be presented. It will be important in the feasibility study to ensure diversity among the recruited participants if possible, capturing people from diverse cultural backgrounds and the type of dementia (which can impact their ability to engage with physical activity). A further limitation was the inability to bring the care partners together into a group setting due to timing constraints, but individual interviews potentially allowed them the freedom to share their views and opinions without fear of judgement and potential power differentials within a group setting. Finally, this study was conducted within a defined geographical location which may limit the generalisability of the findings. However, a positive aspect of this approach is that it allowed us to readily scope the physical activity referrers and providers in the region.

## 5. Conclusions

This study brought together the voices of care partners of, and multiple professional stakeholders working with, people living with dementia toward a common understanding of how best to engage people living with dementia in physical activity. Three personas were developed, which will provide clear guidance for health and care professionals when considering capacity, opportunity and motivation for physical activities. This study has provided the necessary framework from which to develop and test training and resources, with the next stage of intervention (a feasibility trial) currently underway to engage people living with dementia in physical activity options.

## Figures and Tables

**Figure 1 nursrep-15-00002-f001:**
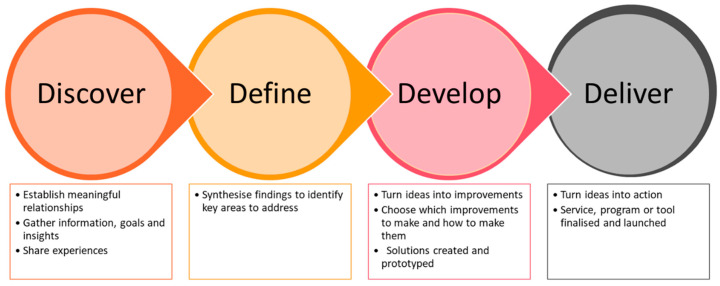
Adapted co-design framework.

**Table 1 nursrep-15-00002-t001:** Preliminary personas from co-design session 1.

**Individual Setting**	**Group Setting (Not So Confident/Social)**	**Group Setting (Confident and Social)**
Suitable for people with dementia who require support: Building confidence for group settings; Trying new things; Addressing safety concerns; With a simplified program to begin with at home; 1:1 soon after diagnosis; Trialling hybrid sessions of in-person sessions and online.	Suitable for people with dementia who prefer the following: A quieter space; Less stimulation (e.g., lighting, noise levels); Familiar people and staff; A buddy for support—informal care partner, volunteer, friend, personal care worker; To be matched to a group that meets their level of ability; Building on familiar activities.	Suitable for people with dementia who are comfortable with the following: A larger group; More complex activities; Previously familiar activities; Music that is motivating; Attending on their own; A physical and mental challenge.

**Table 2 nursrep-15-00002-t002:** Challenges and opportunities for referral pathways.

**Referral Pathway Challenges**	
1. Limited resources of health and care professionals	Referral processes can be complex, time consuming and onerous Need to be in for the ‘long haul’ of the person’s ever-changing dementia process
2. Limited knowledge of health and care professionals, people with dementia and their care partners	Professionals supporting the dementia journey tend to focus on cognitive aspects, with limited attention paid to physical needs Limited feedback loop between referrer and physical activity provider Limited knowledge in readiness for behaviour change related to physical activity, impacting ability to engage people with dementia at the ‘right time’ Limited knowledge of physical activity benefits, supporting both cognitive and physical health
3. Access constraints	Access to private or public transport Fee for service costs and/or aged care funding constraints Reliance on a buddy can be prohibitive for the person with dementia and/or their care partner Limited access to physical activity options that are deemed ‘dementia friendly’
4. Perceived person-related factors	Power imbalance between health and care professionals and the person with dementia/care partner—potential for a compliant approach rather than shared decision making Lack of confidence of the person with dementia and/or care partner in physical activity options to suit their needs and preferences Timing of referral competes with other pressing priorities (e.g., family matters, elective surgery) Changing physical or cognitive circumstances that require attention
**Referral Pathway Opportunities**	
1. Champions	Peer support and/or coaching Formal and informal care partners to encourage and support engagement and participation
2. Streamline referral processes	Minimise referral ‘rules’ to ease the administrative burden Integration of services, with transparent communication between stakeholders Establish a feedback loop between referrers and physical activity providers to ensure best outcomes for the person with dementia and care partner
3. Recognition of triggers for reducing capability and motivation	A change in, or additional, medical conditions Physical decline/increasing frailty Increasing social isolation
4. Beliefs and attitudes of health and care professionals, people with dementia and care partners	Case studies/stories to help shift attitudes towards positive aspects of physical activity Objective outcome measures to support physical changes and benefits Promoting the link between physical and cognitive benefits
5. Communication	Promotional material Dementia-friendly services Positive ageing messages

**Table 3 nursrep-15-00002-t003:** Training program details.

**Module**	**Description and Objectives**
1. An introduction to dementia	This module provides basic information about dementia and describes its common symptoms and causes. Objectives: To describe dementia, its common signs and symptoms To describe the risk factors and ways to reduce risk for dementia To highlight some of the ways to slow the progression of dementia
2. Changes in physical function and benefits of physical activity for people with dementia	This module provides information about the changes in physical function in dementia and the benefits of physical activity participation for people living with dementia. Objectives: To describe changes in physical function and physical activity participation in people with dementia To describe the risk of falls in people with dementia To present evidence for the benefits of exercise and physical activity for people with dementia To describe strategies, aids and technology to support physical function for people living with dementia
3. Engaging people with dementia in exercise programs and/or physical activities	This module describes some common barriers to exercise and/or physical activity participation. It provides some practical strategies that could be used to engage people with dementia to learn and participate in exercise programs and/or physical activities. Objectives: To describe the common barriers to exercise and/or physical activity participation for people with dementia To present a decision support tool for exercise and/or physical activity providers to discuss with the person with dementia and their carers to decide on suitable exercise and/or physical activity options To provide tips for engaging people with dementia to participate in exercises and/or physical activities (including the COM-B framework for behavioural change strategies To present some learning strategies to help people with dementia to participate in exercises and/or physical activities (including strategies to support communication with people with dementia)
4. Environmental design factors for dementia-friendly exercise facilities or environments	This module provides an overview of environmental design principles and features for dementia-friendly environments. Objectives: To understand the values, principles and design considerations of dementia-friendly environments To apply these principles when thinking about facilities in which to provide exercise programs in the future
5. Referral, funding options and resources for people with dementia	This module describes the current referral pathways for older people, including options to consider for people with dementia, to participate in physical activity programs. It provides a summary of the current physical activity programs and useful resources in the Frankston and Mornington Peninsula regions and discusses some of the funding options for physical activity program participation. Objectives: To describe the current referral pathways for older people, including people with dementia, to participate in physical activity programs in the Frankston and Mornington Peninsula regions To discuss the challenges and opportunities in the current referral pathways To summarise the current physical activity programs for older people in the Frankston and Mornington Peninsula regions that could be considered for people with dementia To provide useful resources for finding and accessing (e.g., transport) to local physical activity programs To discuss some funding options for physical activity programs

## Data Availability

The data that support the findings of this study are not publicly available due to privacy restrictions but are available from the corresponding author upon reasonable request.

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
