# Peer review of "‘Changing the Focus’*: Co-Design of a Novel Approach for Engaging People with Dementia in Physical Activity"

_nursrep, 2024, doi:10.3390/nursrep15010002_

Round 1
Reviewer 1 Report
Comments and Suggestions for Authors
The study gives relevant information about engaging people suffering from dementia in physical activity using the co-design method. This method gives a new way of gathering data from several stakeholders, as specially the end-users, patients’ care-partners. Unfortunately, patients were not participated despite several attempts in the study, which is understandable due to the nature of the disease. The gap in knowledge is clearly identified using several references. The amount of references is good as well as the references are mostly quite new and the references are relevant. However, some old references exist. The manuscript doesn’t include inappropriate self-citations. The co-design process has been described clearly. Some references of the method have been presented, but the description of the method is missing. The manuscript includes several tables which are clear and informative. The ethical considerations are clearly presented. In the discussion the authors present the findings and compare the results with previous studies thoroughly. The statements and conclusions are coherent, and they are supported by the listed citations.
Specific comments:
Line 56: Please specify that. What is the conclusion of the study?
Line 107: Please give a brief description of the method you used and also justify why Phase 2 included both Define and Develop.
Line 143: As patients with dementia did not participate in the study (Line 229), is it a good idea to mention it here?
Line 198: The analysis process is missing. How did the authors proceed from the straight quotes to the themes?
Line 279: Please write whole sentences instead of using bullet points or use a table. The font is different than in other parts of the text.
Line 352: Is this listing necessary as the same content is included in the table 3? Please describe the content in sentences.
Author Response
I have uploaded a table as part of the cover letter providing a point-by-point response to reviewer comments.

Reviewer 2 Report
Comments and Suggestions for Authors
Thank you for this well-written manuscript. I would like to specifically commend you for your clear and detailed methods section. I have a few queries/ comments in relation to the paper, specifically the findings/ discussion sections.
1. There are some font/ formatting inconsistencies.
2. Findings/ discussion: I recognise the sample was small, but nonetheless it would be useful to have some information on the care partners in terms of socio-demographics and specific dementia diagnosis. This leads me to my other comment re cultural considerations and diagnosis specificity for these type of interventions. I appreciate that the sample was small and despite efforts made, no one living with a dementia diagnosis could be recruited. This naturally limits the opportunity for a diverse sample, yet it would be good to see some discussion in terms of cultural considerations and its importance and implications for the feasibility study.
Furthermore, we know that dementia diversity matters (the study's inclusion criteria focuses on cognitive impairment as assessed by the MMSE rather specific dementia diagnosis) since different dementia diagnoses impact peoples' ability to participate in physical activity differently. For example posterior cortical atrophy and Lewy body dementia affect individual's vision and mobility in ways other types of dementia may do to a lesser extent. Given reference made to "memory-led" dementia, it seems the authors primarily focus on Alzheimer's disease. It would be useful to either be specific in the inclusion criteria if the intervention is solely aimed at people living with AD or discuss implications for other, rarer types of dementias. It would be important to discuss this point in view of the future work proposed.
Author Response
I have responded to the reviewer comments as part of the cover letter.

Reviewer 3 Report
Comments and Suggestions for Authors
Thank you for the opportunity to review this paper. It presents an interesting perspective on a well-known challenge: sustaining and continuing physical exercise programs, which is difficult not only for people with dementia but for the general population as well. The results are well-presented. However, addressing the challenges of participant retention and program continuation beyond the intervention remains a significant hurdle.
Some comments:
In general, I was continuously wondering how the proposed exercise program is different to other exercise program. Authors need to mention that the aim of the co-design process is to create a guidelines to enhance opportunity and motivation for physical program rather than developing the program.
Abstract: Method: it is not clear who the stakeholders are in the study. also it is not clear (as well as in Methodology 2.2, ) if the care-partners are non-paid family members and/or paid carers, aged-care providers.
Methods: study design: it is not clear how Personal is different to Thematic analysis?
Results: the authors mentioned line 229 “Despite numerous recruitment attempts, no participants with dementia were recruited.” Their method of numerous attempts in the methods is not convincing to recruit people with dementia to give their voice in the exercise program.
Findings from Session 3 for stakeholders workshop is not reported.
Line 301-303: both care-partner/friend and Formal care-partners are mentioned here, but not in the methods or results. Further, it is unclear of the perspectives are from the formal care-partners themselves or friend care-partner’s perception.
4.2.1 Participant’s direct quotes will be appropriate to support and elaborate the persona.
Tables: inconsistent text style across the tables.
Discussion: lines 389-393 the sentences do not follow the previous paragraph. It seems to be ideal to move it to the introduction.
Author Response
I have responded to the reviewer comments in the cover letter.
